# Comparative brain-wide mapping of ketamine- and isoflurane-activated nuclei and functional networks in the mouse brain

Yue Hu[1†], Wenjie Du[1†], Jiangtao Qi[1†], Huoqing Luo[2], Zhao Zhang[1], Mengqiang Luo[1]*, Yingwei Wang[1]*

[1]Department of Anesthesiology, Huashan Hospital, Fudan University, Shanghai, China; [2]School of Life Science and Technology, ShanghaiTech University, Shanghai, China

*For correspondence:
luomq16@fudan.edu.cn (ML);
wangyw@fudan.edu.cn (YW)

[†]These authors contributed equally to this work

Competing interest: The authors declare that no competing interests exist.

**Abstract** Ketamine (KET) and isoflurane (ISO) are two widely used general anesthetics, yet their distinct and shared neurophysiological mechanisms remain elusive. In this study, we conducted a comparative analysis of the effects of KET and ISO on c-Fos expression across the mouse brain, utilizing hierarchical clustering and c-Fos-based functional network analysis to evaluate the responses of individual brain regions to each anesthetic. Our findings reveal that KET activates a wide range of brain regions, notably in the cortical and subcortical nuclei involved in sensory, motor, emotional, and reward processing, with the temporal association areas (TEa) as a strong hub, suggesting a top-down mechanism affecting consciousness by primarily targeting higher order cortical networks. In contrast, ISO predominantly influences brain regions in the hypothalamus, impacting neuroendocrine control, autonomic function, and homeostasis, with the locus coeruleus (LC) as a connector hub, indicating a bottom-up mechanism in anesthetic-induced unconsciousness. KET and ISO both activate brain areas involved in sensory processing, memory and cognition, reward and motivation, as well as autonomic and homeostatic control, highlighting their shared effects on various neural pathways. In conclusion, our results highlight the distinct but overlapping effects of KET and ISO, enriching our understanding of the mechanisms underlying general anesthesia.

## eLife assessment

This **important** study used single-cell whole-brain imaging of the immediate early gene Fos to identify the brain areas recruited by two anesthetics, ketamine and isoflurane. The utilization of a custom software package to align and analyze brain images for c-Fos positive cells stands out as an impressive component of the approach. The results provide **solid** evidence that these anesthetics might induce anesthesia via different brain regions and pathways, and raw fos showed shared and distinct activation patterns after ketamine- v. isoflurane-based anesthesia. Though differences could also be due, as the authors note, to differences in dose and route of administration. This paper may be of interest to preclinical and clinical scientists working with anesthetic and dissociative drugs.

## Introduction

Despite considerable investigation into molecular targets, neural circuits, and functional connectivity associated with various anesthetics, our understanding of their effects on overall brain activity

continues to be limited and incomplete (*Hemmings et al., 2019*). At the molecular level, ketamine (KET) modulates neuronal excitability predominantly through N-methyl-D-aspartate (NMDA) receptors, whereas isoflurane (ISO) primarily affects gamma-aminobutyric acid type A (GABAa) receptors. Their distinct mechanisms of action can each contribute to the loss of consciousness. (*Franks, 2008*). In systems neuroscience, the neural mechanisms of anesthetic induced unconsciousness involve both top-down and bottom-up processes (*Mashour, 2014*; *Mashour and Hudetz, 2017*). As evidenced by in vivo electrophysiology or functional magnetic resonance imaging (fMRI) studies, the top-down paradigm demonstrates that anesthetics induce unconsciousness by disrupting corticocortical and corticothalamic circuits responsible for neural information integration, while peripheral sensory information can still be conveyed to the primary sensory cortex (*Schroeder et al., 2016*; *Lee et al., 2013*). The bottom-up approach, exemplified by ISO, involves the activation of sleep-promoting nuclei like ventrolateral preoptic nucleus (VLPO) and inhibition of arousal centers in the brainstem and diencephalon, supporting the shared circuits of sleep and anesthesia (*Moore et al., 2012*; *Nelson et al., 2002*). However, the limited spatial resolution of fMRI studies and the inability of electroencephalography (EEG) to capture specific brainstem nuclei significantly hinder the acquisition of comprehensive whole-brain information. Although a substantial body of knowledge has been gathered, our understanding of the reciprocal responses among different brain regions during general anesthesia remains relatively sparse and fragmented. To bridge these gaps, further investigation using advanced techniques that can capture whole-brain dynamics is needed to elucidate the complex interactions and shared mechanisms between various anesthetics.

Neuronal extracellular stimulation typically results in the elevation of adenosine 3',5'-cyclic monophosphate (cAMP) levels and calcium influx, ultimately leading to the upregulation of immediate early genes (IEGs) such as c-fos (*Yap and Greenberg, 2018*; *Morgan and Curran, 1989*). The translation product of c-fos, c-Fos protein, offers single-cell spatial resolution and has been utilized as a biomarker to identify anesthetic-activated brain regions (*Zhang et al., 2022*). Previous investigations of c-Fos expression throughout the brain demonstrated that GABAergic agents inhibited cortical activity while concurrently activating subcortical brain regions, including VLPO, median preoptic nucleus (MEPO), lateral septal nucleus (LS), Edinger-Westphal nucleus (EW), and locus coeruleus (LC; *Smith et al., 2016*; *Lu et al., 2008*; *Yatziv et al., 2020*; *Han et al., 2014*). In contrast, KET was shown to provoke wake-like c-Fos expression and intense augmentation of c-Fos expression in various brain regions at clinical dosages (75–100 mg/kg; *Lu et al., 2008*). It is important to note that these experiments administered KET at lights-on and GABAa receptor agonists at lights-off, potentially introducing circadian influences for direct comparison of KET and ISO. Moreover, it has been revealed that the state of general anesthesia is not determined by activity in individual brain areas but emerges as a global change within the brain. This change involves the activation of lateral habenular nucleus (LHb), VLPO, supraoptic nucleus (SO), and central amygdaloid nucleus (CeA), which are essential for anesthetic induced sedation, unconsciousness, or analgesia (*Moore et al., 2012*; *Gelegen et al., 2018*; *Hua et al., 2020*; *Jiang-Xie et al., 2019*). However, comprehensive brain-wide mapping and comparison of distinct anesthetic (KET and ISO) activated nuclei at the cellular level has not been fully elucidated.

In this study, we initially investigated the distribution of nuclei activated by KET and ISO across 984 brain regions. This was accomplished through immunochemical labeling and utilizing a customized MATLAB software package, which aided in signal detection and alignment with the Allen Mouse Brain Atlas reference (*Ma et al., 2021*). Subsequently, we analyzed whole-brain c-Fos expression elicited by KET and ISO using hierarchical clustering, and assessed inter-regional correlations by calculating the covariance across subjects for each brain region pair. Regions with significantly positive correlations were identified to construct functional networks, and graph theory-based methods were applied to identify hub nodes. Our findings revealed distinct yet overlapping whole-brain activation patterns for KET and ISO.

## Results

### Identification of brain regions activated by KET

To investigate the pattern of c-Fos expression throughout the brain, either 1.5% ISO was continuously ventilated or 100 mg/kg KET was given 90 min before harvesting (*Figure 1A*). Sample collection for all mice was uniformly conducted at 14:30 (ZT7.5), and the c-Fos labeling and imaging procedures

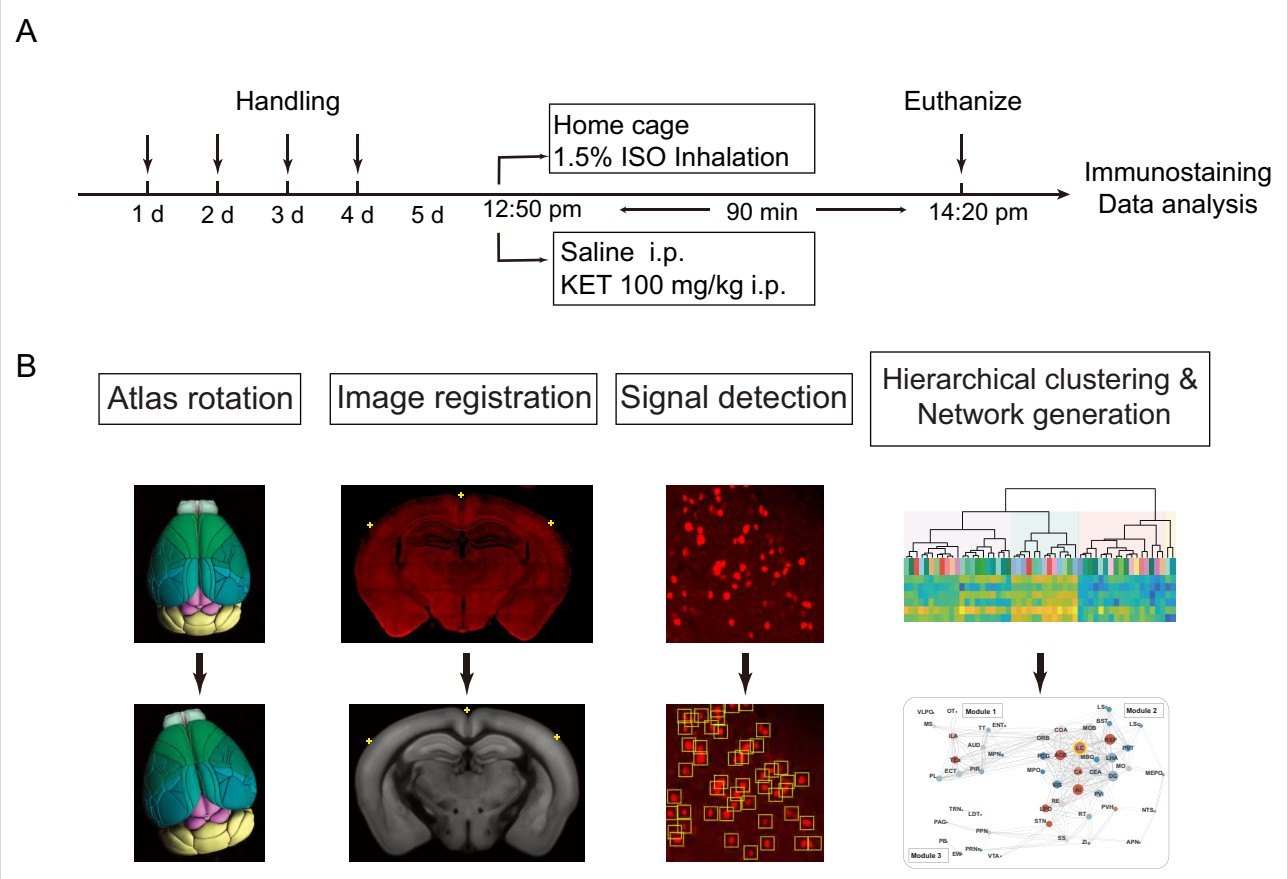

**Figure 1.** Experimental design and workflow for data analysis. (**A**) Experimental timeline for anesthetic exposure. Mice were handled for 4 days, subsequently exposed to KET (100 mg/kg, i.p.) or 1.5% ISO for 90 min, and then euthanized for immunostaining and data analysis. (**B**) Steps for data processing. Example of brain section registration to a corresponding coronal section from the Allen Brain Atlas (***Allen Institute for Brain Science, 2011***). For Atlas rotation, the Allen reference atlas was rotated to mimic the slice angle of the experimental brain. Image registration maps the original coronal image (upper panel) to the corresponding Allen Mouse Brain Atlas (lower panel). The registration module applies several geometric transformations (translation, rotation, and scaling) to optimize the matching of the original image to the anatomical structures. Fluorescence signals detected in the original image were projected onto the Allen Mouse Brain Atlas for quantification. Finally, the processed data undergo hierarchical clustering and network analysis to investigate the patterns of c-Fos expression and central network nodes.

The online version of this article includes the following figure supplement(s) for figure 1:

**Figure supplement 1.** EEG and EMG power change after each treatment.

were performed using consistent parameters across all experiments. To quantitatively assess brain states during anesthesia, we utilized EEG and electromyography (EMG) recordings. The depth of anesthesia was evaluated based on the power ratio of the EEG signals within the delta (0.5–4 Hz) and theta (6–10 Hz) frequency bands, combined with the changes in the EMG (***Luo et al., 2023***). Our results indicated that the administration of either KET or ISO elevated the EEG delta/theta power ratio while reducing EMG power (***Figure 1—figure supplement 1***), suggesting that each drug induced a loss of consciousness. The raw images of brain slices were aligned to the Allen Mouse Brain Atlas, followed by signal detection and analysis (***Figure 1B***). Building upon the framework established in previous literature, our study first categorizes the whole brain of mice into 53 distinct subregions (***Do et al., 2016***). To assess the brain regions activated by KET and ISO in comparison to their respective controls, Z-scores were calculated for a statistical comparison of c-Fos expression differences. This process entailed normalizing the mean expression difference against the standard deviation of the control group. Our findings revealed that KET elicited significant activation in almost all 53 brain regions when compared to the saline group. Conversely, ISO showed substantial activation across various regions, excluding the cortical areas, when compared to the home cage group (***Figure 2—figure supplement 1***).

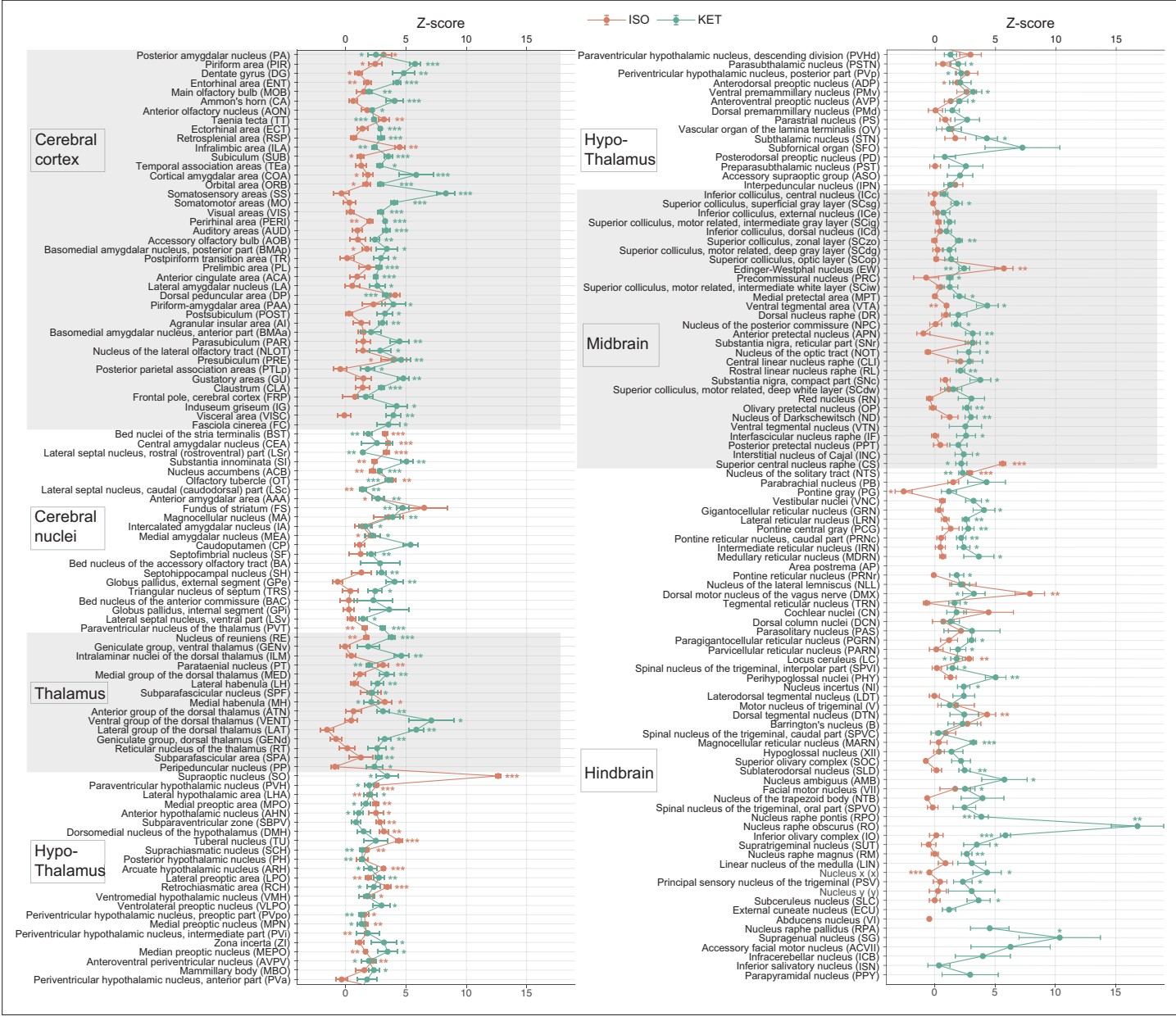

**Figure 2.** KET and ISO induced c-Fos expression relative to their respective control group across 201 distinct brain regions. Z-scores represent the normalized c-Fos expression in the KET and ISO groups, calculated against the mean and standard deviation from their respective control groups. Statistical analysis involved the comparison of Z-scores to a null distribution with a zero mean and adjustment for multiple comparisons using the Benjamini–Hochberg method at a 5% false discovery rate (*p<0.05, **p<0.01, ***p<0.001). n=6, 6, 8, 6 for the home cage, ISO, saline, and KET, respectively. Missing values resulted from zero standard deviations in control groups. Brain regions are categorized into major anatomical subdivisions, as shown on the left side of the graph.

The online version of this article includes the following source data and figure supplement(s) for figure 2:

**Source data 1.** Z-scores of c-Fos expression in each brain region following KET and ISO treatments.

**Figure supplement 1.** KET and ISO induced c-Fos expression relative to their respective control group across 53 distinct brain regions.

To enable a more detailed examination and facilitate clearer differentiation and comparison of the effects caused by KET and ISO, we subdivided the 53 brain regions into 201 distinct areas. This approach, guided by the standard mouse atlas available at http://atlas.brain-map.org/atlas, allowed for an in-depth analysis of the responses in various brain regions. Compared to the saline group, KET activated 141 out of a total of 201 brain regions (*Figure 2*). To further identify the brain regions that are most significantly affected by KET, we calculated Cohen's d for each region to quantify the

magnitude of activation and subsequently focused on those regions that had a corrected p-value below 0.05 and effect size in the top 40% (*Figure 3*, *Figure 3—figure supplement 1*). Our findings were in line with previous studies: numerous cortical regions associated with somatosensory, auditory, visual, and movement were activated (*Figure 3—source data 1*). Notably, KET activated sensory areas, such as visual areas (VIS), auditory areas (AUD), somatosensory areas (SS), gustatory areas (GU), and higher-order cortical areas, including the anterior cingulate area (ACA), temporal association areas (TEa), retrosplenial area (RSP), and dorsal peduncular area (DP). This contrasts with the usual inhibiting effect of anesthetics on these neural circuits and suggests the unique ability of KET to sustain certain neuronal circuits in an active state (*Brown et al., 2011*). Furthermore, its impact on the olfactory system, involving key structures such as the anterior olfactory nucleus (AON), main olfactory bulb (MOB), olfactory tubercle (OT), perirhinal area (PERI), piriform area (PIR), and ectorhinal area (ECT), was distinct compared to other types of general anesthetics (*Zhao et al., 2020*). Moreover, KET affected emotional and reward processing regions such as the nucleus accumbens (ACB; *Floresco, 2015*), orbital area (ORB; *Rolls et al., 2020*), infralimbic area (ILA), and prelimbic area (PL; *Pfarr et al., 2018*), suggesting an impact on the circuitry responsible for modulating reward and affective states. The thalamus, including the paraventricular nucleus of the thalamus (PVT) (*Ren et al., 2018*) and lateral group of the dorsal thalamus (LAT) (*Suzuki and Larkum, 2020*), which are vital in arousal regulation, also showed enhanced activity under the influence of KET. Hindbrain regions under KET anesthesia, including the magnocellular reticular nucleus (MARN), paragigantocellular reticular nucleus (PGRN), lateral reticular nucleus (LRN), nucleus raphe obscurus (RO), and sublaterodorsal nucleus (SLD), showed notable activity. In particular, the activation in regions like PGRN and SLD suggests that KET modulates brain regions critical for the regulation of rapid eye movement (REM) sleep (*Goutagny et al., 2008*). The activation of diverse regions such as the fundus of striatum (FS), substantia innominata (SI), and agranular insular area (AI), underscores that the effects of KET are not confined to classical neural pathways, but rather encompass a wide array of brain networks. Overall, KET induces marked activation across various brain regions, encompassing not only higher order cortical areas but also those involved in sensory processing, emotional regulation, reward systems, and arousal mechanisms, thereby highlighting its distinct and widespread influence on diverse neural networks.

## Identification of brain regions activated by ISO

To identify brain regions activated by ISO, we first reviewed nuclei previously reported to be activated by ISO. These encompass the PIR in the cortex, the lateral septal nucleus (LS), and the central amygdaloid nucleus (CEA) in the striatum. Additionally, activations were observed in hypothalamic regions such as the SO, VLPO, and MPO. Activations were also observed in the EW, the nucleus of the solitary tract (NTS), LC, the ventral group of the dorsal thalamus (VeN), and the area postrema (AP; *Figure 4—source data 1*). Building upon these established findings, our study further explored the effects of ISO. Using the same criteria applied to KET, which involved selecting regions with Cohen's d values in the top 40% of significantly activated areas from *Figure 2*, we identified 32 key brain regions impacted by ISO (*Figure 4*, *Figure 4—figure supplement 1*). Our findings not only confirmed the activation of known regions like CeA, SO, NTS, and LC, but also broadened the scope of understanding regarding the effects of ISO, revealing its impact on several previously unexplored areas. This included the paraventricular hypothalamic nucleus (PVH), arcuate hypothalamic nucleus (ARH), lateral hypothalamic area (LHA), tuberal nucleus (TU), median preoptic nucleus (MEPO), lateral preoptic area (LPO), retrochiasmatic area (RCH), periventricular hypothalamic nucleus intermediate part (PVi), dorsomedial nucleus of the hypothalamus (DMH), and anteroventral periventricular nucleus (AVPV). The activation of these regions suggested a substantial influence of ISO on hypothalamic regulation, including neuroendocrine control, autonomic function, and homeostasis (*Gu et al., 1999*; *Adamantidis and de Lecea, 2023*). Additionally, ISO affected the areas governing circadian rhythms and sleep-wake regulation, as indicated by activation in the suprachiasmatic nucleus (SCH) and subparaventricular zone (SBPV; *Lu et al., 2001*). Moreover, sensory processing and reward circuitry, including the OT (*Giessel and Datta, 2014*), SI (*Cui et al., 2022*), and ACB (*Zhou et al., 2022*), demonstrated significant activation. Activation of the perirhinal area (PERI; *Yang and Naya, 2023*), and nucleus of reuniens (RE; *Huang et al., 2021*) pointed to a potential impact on cognitive processes and memory systems. Our study also revealed ISO activates the EW, a finding that aligned with observations under sevoflurane anesthesia

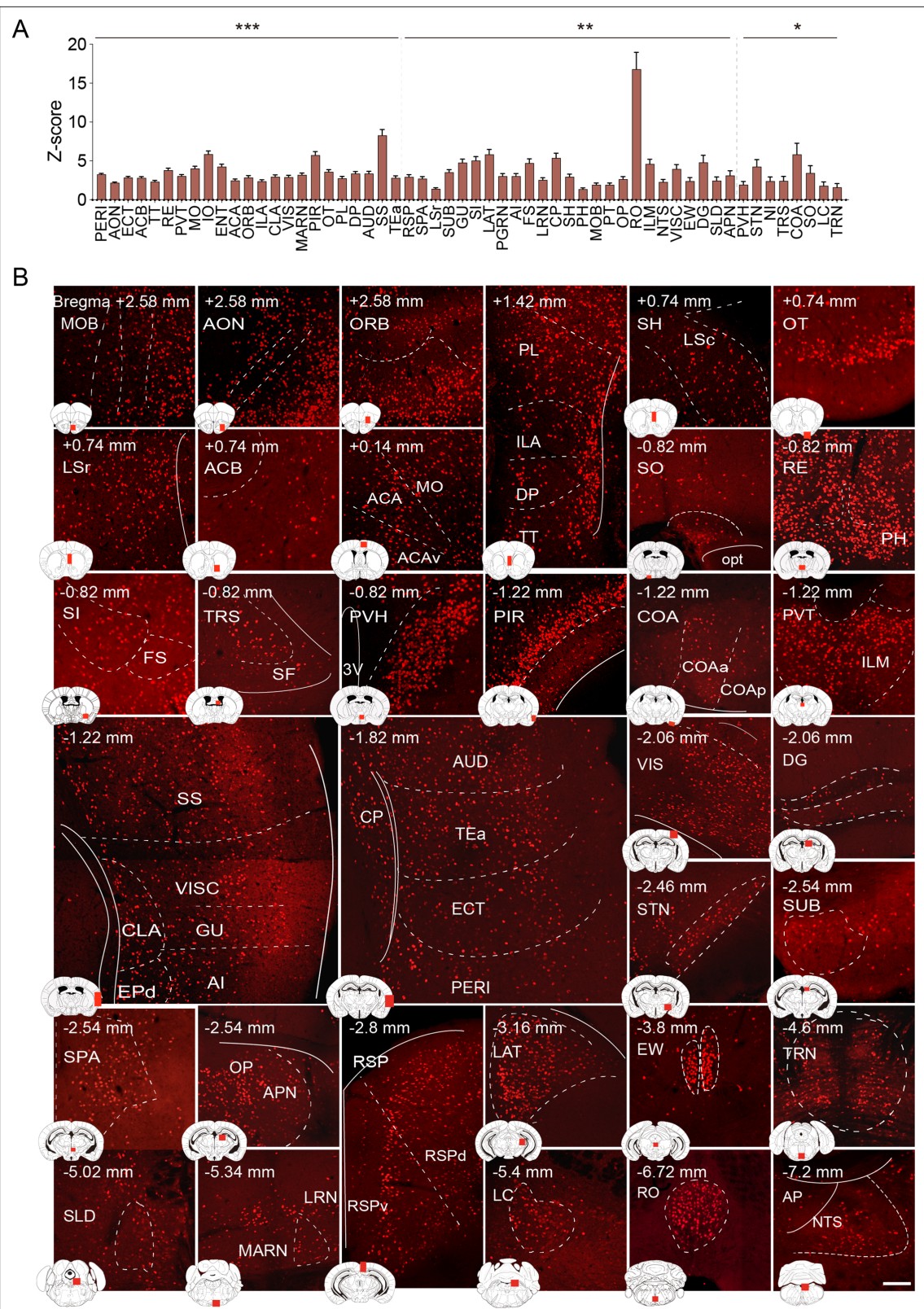

**Figure 3.** Brain regions exhibiting significant activation by KET. (**A**) Fifty-five brain regions exhibited significant KET activation. These were chosen from the 201 regions analyzed in *Figure 2*, focusing on the top 40% ranked by effect size among those with corrected p values less than 0.05. Data are presented as mean ± SEM, with p-values adjusted for multiple comparisons (*p<0.05, **p<0.01, ***p<0.001). (**B**) Representative immunohistochemical staining of brain regions identified in **A**, with control group staining available in *Figure 3—figure supplement 1*. Scale bar: 200 μm.

*Figure 3 continued on next page*

*Figure 3 continued*

The online version of this article includes the following source data and figure supplement(s) for figure 3:

**Source data 1.** Summary of prior studies on KET activated brain regions identified via c-Fos immunostaining.

**Figure supplement 1.** c-Fos expression in distinct brain regions following saline administration.

(*Yi et al., 2023*). The bed nuclei of the stria terminalis (BST), associated with stress responses and emotional regulation, exhibited significant activation (*Daniel and Rainnie, 2016*). In summary, our findings suggest that ISO predominantly influences the hypothalamus, a key area for neuroendocrine regulation, autonomic control, and homeostasis, while also influencing brain functions associated with sensory processing, reward mechanisms, stress response, and cognitive functions.

The co-activation of multiple brain regions by KET and ISO indicates that they have overlapping effects on brain functions. Examples of these effects include impacts on sensory processing, as evidenced by the activation of the PIR, ENT (*Chapuis et al., 2013*), and OT (*Giessel and Datta, 2014*), pointing to changes in sensory perception typical of anesthetics. Memory and cognitive functions are influenced, as indicated by the activation of the subiculum (SUB; *Roy et al., 2017*), dentate gyrus (DG; *Sun et al., 2020*), and RE (*Huang et al., 2021*). The reward and motivational systems are engaged, involving the ACB and ventral tegmental area (VTA), signaling the modulation of reward pathways (*Al-Hasani et al., 2021*). Autonomic and homeostatic control are also affected, as shown by areas like the lateral hypothalamic area (LHA) (*Mickelsen et al., 2019*) and medial preoptic area (MPO) (*McGinty and Szymusiak, 1990*), emphasizing effects on functions such as feeding and thermoregulation. Stress and arousal responses are impacted through the activation of the paraventricular hypothalamic nucleus (PVH) (*Rasiah et al., 2023*; *Islam et al., 2022*) and LC (*Ross and Van Bockstaele, 2020*). This broad activation pattern highlights the overlap in drug effects and the complexity of brain networks in anesthesia.

## Comparison of c-Fos activation patterns in response to KET and ISO

To reveal common activation patterns and potential associations among the brain regions responding to KET and ISO, hierarchical clustering was employed on 53 brain regions as delineated in *Figure 2—figure supplement 1*. We normalized the c-Fos densities in each of the 53 regions to the average values of that region in the control group and then log-transformed the data. Hierarchical clustering of these values revealed distinct patterns of drug-induced changes (*Figure 5A*), with both KET and ISO showing optimal clustering at a 0.5 cut-off value, as confirmed by high silhouette coefficients (*Figure 5B*). Concurrently, we computed the number of clusters formed at each potential cut-off value and found that a 0.5 threshold resulted in four distinct clusters for both KET and ISO treatments (*Figure 5C*). After hierarchical clustering, we preserved the sequence of brain regions and replaced log relative c-Fos densities with Z values, calculated by dividing the log relative c-Fos density in each brain region by the standard error, for both KET and ISO conditions (*Figure 5E*). KET administration elicited a significant activation in a broad array of brain regions, with cluster 4 displaying the most substantial upregulation in expression. This cluster incorporates cortical areas, including the visceral area (VISC), GU, ORB, somatomotor areas (MO), ACA, and SS, and cerebral nuclei that are integral for motor coordination and motivational behaviors, specifically, the dorsolateral and ventrolateral pallidum (PALd, PALv) and striatum (STRd, STRv) (*Grillner et al., 2005*, *Figure 5D and E*). No significantly inhibited brain regions were detected in the ISO group (as no Z value fell below –2); clusters 3 and 4 expression levels were similar to or lower than those in the control group, with cluster 3 having a mean Z value of 0.27 and cluster 4 a mean Z value of –1.1, as shown in *Figure 5D and E*. The most pronounced upregulation of ISO-induced c-Fos expression was observed in the second cluster (mean Z value = 4.7), which includes brain regions responsible for a variety of functions: the periventricular zone (PVZ) and lateral zone (LZ) of the hypothalamus, vital for endocrine and autonomic system regulation *Gu et al., 1999*; the ventral striatum (STRv), pallidum (PAL), and lateral septal complex (LSX) of the striatum, key areas in reward and motivation *Cox and Witten, 2019*; the taenia tecta (TT), PIR, essential for olfactory processing (*Bekkers and Suzuki, 2013*; *Kataoka et al., 2020*). This collective upregulation in cluster 2, indicates a consistent activation pattern and drug response among diverse functional brain regions. It underscores the impact of ISO on hypothalamic regulation, reward mechanisms, and olfactory processing, with marked activation observed in the hypothalamic and striatal

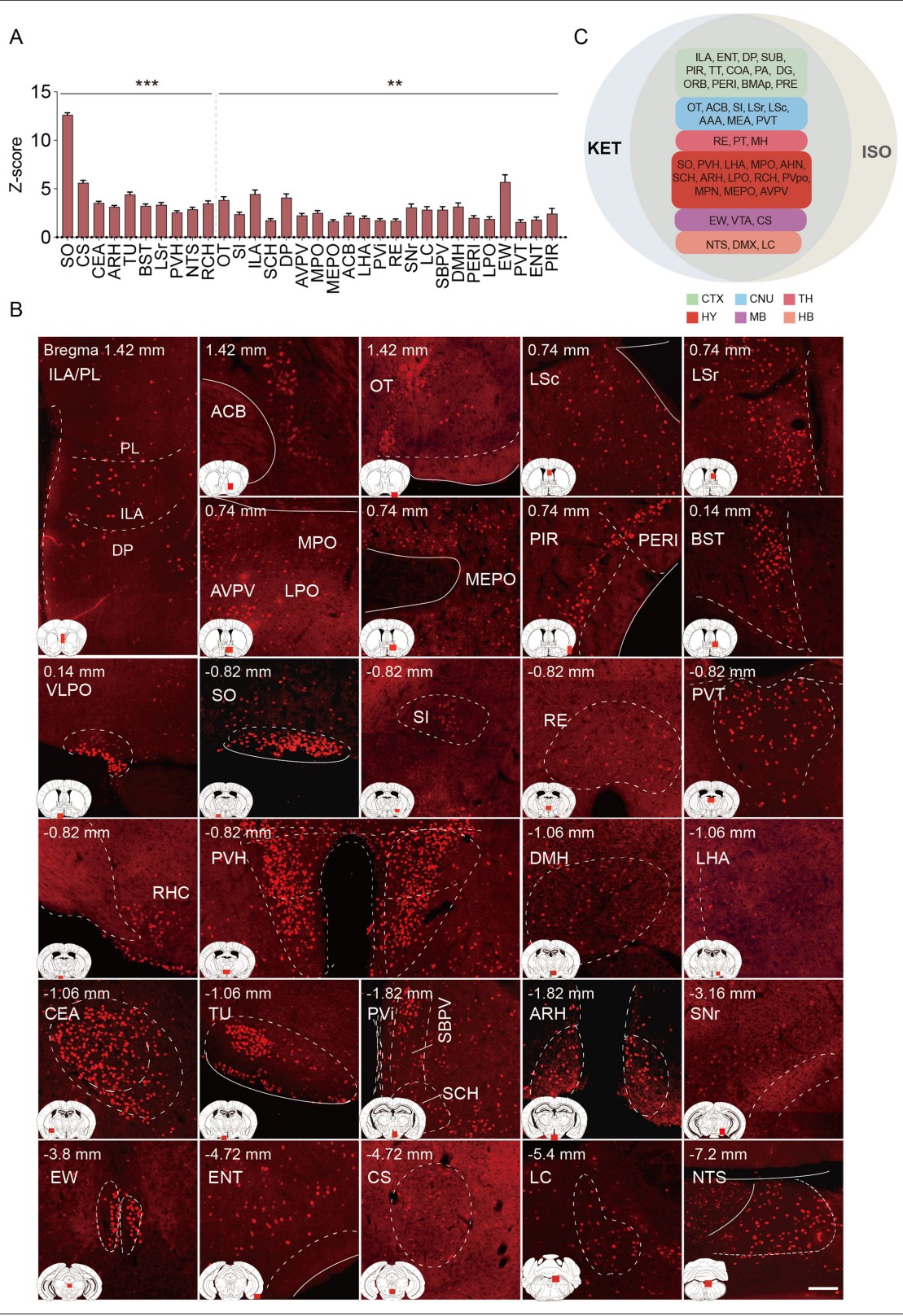

**Figure 4.** Brain regions exhibiting significant activation by ISO. (**A**) Brain regions significantly activated by ISO were initially identified using a corrected p-value below 0.05. From these, the top 40% in effect size (Cohen's d) were further selected, resulting in 32 key areas. p-values are adjusted for multiple comparisons (**p<0.01, ***p<0.001). (**B**) Representative immunohistochemical staining of brain regions identified in **A**. Control group staining is available in *Figure 4—figure supplement 1*. Scale bar: 200 μm. Scale bar: 200 μm. (**C**) A Venn diagram displays 43 brain regions co-activated

*Figure 4 continued on next page*

*Figure 4 continued*

by KET and ISO, identified by the adjusted p-values (p<0.05) for both KET and ISO. CTX: cerebral cortex; CNU: cerebral nuclei; TH: thalamus; HY: hypothalamus; MB: midbrain; HB: hindbrain.

The online version of this article includes the following source data and figure supplement(s) for figure 4:

**Source data 1.** Summary of prior studies on brain regions activated by ISO, as detected through c-Fos immunostaining.

**Figure supplement 1.** c-Fos expression in home cage group.

regions. In contrast, KET induces widespread activation across the brain, with heightened activity in a key cluster that includes cortical and cerebral nuclei governing complex sensory, emotional, and motor processes, thus highlighting its notable influence on higher order functions.

## Similarities and differences in KET and ISO-activated c-Fos brain areas

To achieve a more granular analysis and better discern the differences in responses between KET and ISO, we expanded our study from the initial 53 brain regions to 201 distinct subregions. After log-transforming the c-Fos densities relative to controls, we applied hierarchical clustering to uncover patterns of brain activity (*Figure 6A*). ISO demonstrates a more concentrated influence on brain activity with consistently higher clustering quality and fewer clusters than KET, suggesting a focused impact on specific brain functions (*Figure 6B and C*). *Figure 6E* maintains the order from hierarchical clustering and marks the most activated brain regions in the top 10% of Z values with red boxes and areas of potential suppression with Z values below –2 with white boxes. In the hierarchical clustering analysis of KET exposure, cluster 2 with a mean Z value of –0.01 was similar to the control group in terms of activation, while the other four clusters showed increased activation. Cluster 5, showing the most significant activation, is composed mainly of the cortical and striatal regions crucial for sensory processing, spatial memory, reward, and cognitive functions (*Haber, 2016*). In addition, brain regions within the thalamus and hindbrain, including RE, inferior olivary complex (IO), nucleus raphe obscurus (RO), MARN, and PGRN, also display similar activation patterns. This composition, centered on cortical regions, implies that KET primarily targets higher order brain functions while also influencing fundamental neurological processes through its effect on cerebral nuclei, the thalamus, and the hindbrain. ISO hierarchical clustering identified three distinct clusters, with cluster 3 exhibiting the highest mean Z value of 7.27. This cluster encompasses the SO, VLPO, TU, and CEA (*Figure 6E*). Notably, the SO, VLPO, and CEA have been previously reported to play critical roles in the mechanisms of anesthesia (*Moore et al., 2012*; *Hua et al., 2020*; *Jiang-Xie et al., 2019*). Cluster 1 showed a relative decrease in expression (n=91, mean Z value: –0.54), indicating that nearly half of the brain areas had reduced c-Fos expression compared to the control group. Cluster 2, characterized by overall moderate activation, was particularly significant in the top 10% of its regions based on Z value ranking, as highlighted by the red box (*Figure 6E*). This segment includes key regions such as ARH and PVH, which were also prominently activated as shown in *Figure 4*. It furthermore encompasses cortical areas such as ILA, DP, and TT cortices involved in emotional and reward processing (*Pfarr et al., 2018*; *Kataoka et al., 2020*) and integrates striatal and pallidal regions like OT and ACB and midbrain and hindbrain structures such as AP and NTS. The collective inclusion of these diverse regions, exhibiting pronounced and similar activation patterns, suggests a unified response to ISO, indicative of potential functional connectivity and interaction across these areas. In conclusion, the activation of specific areas such as the SO, VLPO, TU, and CEA by ISO in cluster 3 suggests a targeted influence. In contrast, the extensive engagement of cortical areas by KET in cluster 5, along with its effects on striatal and thalamic regions, underscores a profound and widespread modulation of both higher order and fundamental neural functions.

## Network generation and Hub identification

Previous research has established that general anesthesia is mediated by various brain regions (*Moore et al., 2012*; *Gelegen et al., 2018*; *Hua et al., 2020*; *Jiang-Xie et al., 2019*). The expression of c-Fos serves as an indicator of neuronal activity, providing a single index of activation per region per animal. By examining the covariance of c-Fos expression across animals within each group, we can infer interactions between brain regions and identify functional networks engaged during general anesthesia (*Wheeler et al., 2013*). Initially, 63 brain regions were incorporated, including those significantly

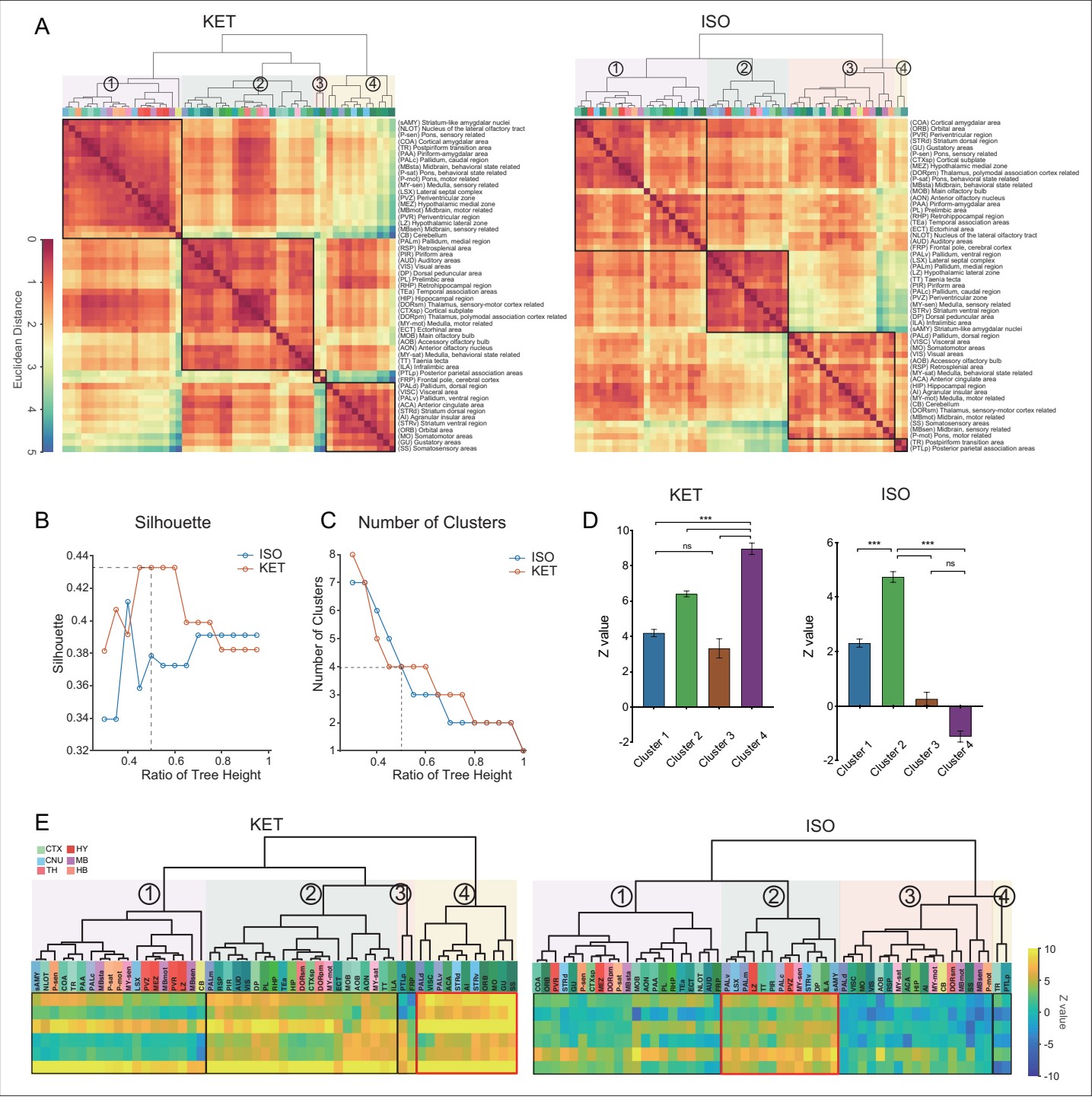

**Figure 5.** Comparison of c-Fos activation patterns in response to KET and ISO. (**A**) Hierarchical clustering of log-relative c-Fos density data for KET and ISO was conducted using the complete linkage method, based on the Euclidean distance matrix. Clusters were identified using a dendrogram cut-off ratio of 0.5, with numerical labels indicating distinct clusters. (**B**) Silhouette values plotted against the dendrogram tree height ratio for KET and ISO reveal higher Silhouette values at a 0.5 ratio (dashed line), suggesting optimal clustering at this level. (**C**) The number of clusters for each treatment condition at different dendrogram tree height ratios is shown, with a cut-off level of 0.5 yielding 4 clusters for both KET and ISO (marked by the dashed line). (**D**) Bar graphs represent the Z values for clusters under both KET and ISO conditions. Data are presented as mean ± SEM. One-way ANOVA with Tukey's post hoc multiple comparisons. ns: no significance; ***$p<0.001$. (**E**) Heatmaps illustrate Z values of the log relative c-Fos expression density in clustered brain regions. The arrangement and abbreviations of the brain regions, along with the numerical labels, correspond to those in **A**. Red box in each heatmap emphasizes the cluster with the highest mean Z value in its respective condition. CTX: cerebral cortex; CNU: cerebral nuclei; TH: thalamus; HY: hypothalamus; MB: midbrain; HB: hindbrain.

*Figure 5 continued on next page*

*Figure 5 continued*

The online version of this article includes the following source data for figure 5:

**Source data 1.** Hierarchical clustering of 53 brain regions based on log-relative c-Fos density responses to KET and ISO treatments, related to *Figure 5A*.

**Source data 2.** Z values of the log relative c-Fos density in 53 brain regions in response to KET and ISO, related to *Figure 5E*.

activated by KET and ISO (as indicated in *Figures 3 and 4*) and others previously associated with general anesthesia and sleep-wakefulness mechanisms (*Leung et al., 2014*). A comprehensive inter-regional correlation for four groups was calculated (*Figure 7A*). Correlation analysis revealed similar network connectivity and densities between the ISO and home cage groups (mean interregional correlations: Fisher Z=–0.018, p=0.98; network densities: 0.10 vs 0.09). In contrast, KET significantly increased network density (0.46 vs 0.06) and mean interregional correlations compared to the saline group (Fisher Z=3.54, p<0.001; *Figure 7—figure supplement 1D*), indicating enhanced interregional correlation with KET administration. Functional networks were segmented into modules based on hierarchical clustering of correlation coefficients (*Figure 7—figure supplement 1A*), with dendrogram cutting at a value of 0.7 for high Silhouette scores (*Figure 7—figure supplement 1B*). This resulted in two modules for the home cage and KET groups, and four for the saline and ISO groups (*Figure 7—figure supplement 1C*). Networks were formed using Pearson's coefficients over 0.82 and significant correlations (p<0.05), yielding a single module for KET and three modules for ISO (*Figure 7B*).

Hubs are nodes that occupy critical central positions within the network and enable the network to function properly. Due to the singular module structure of the KET network and the sparsity of intermodular connections in the home cage and saline networks, the assessment of network hub nodes did not employ within-module degree Z-score and participation coefficients as these measures predominantly underscore the importance of connections within and between modules (*Kimbrough et al., 2020*). The analysis evaluated nodal importance using degree (number of direct connections), betweenness centrality (proportion of shortest network paths passing through a node), and eigen-vector centrality (influence based on the centrality of connected nodes), identifying brain regions that scored highly in all three as hub nodes (*Figure 7—figure supplement 2*). The LC, characterized by its high degree and eigenvector centrality, as well as its notably higher betweenness centrality (*Figure 7—figure supplement 2*) underscores its pivotal role in propagating and integrating neural signals within the ISO-influenced network. In the KET group, the TEa serves as a hub, emphasizing its integrative role in KET-induced dissociative anesthesia. Meanwhile, the periaqueductal gray (PAG) and LSr function as connector hubs in the home cage and saline groups, respectively.

## Discussion

In this study, we conducted a comparative analysis to examine the effects of two general anesthetics, KET and ISO, on c-Fos expression throughout various brain regions. Our results revealed that KET predominantly activates the cerebral cortex, with the TEa emerging as the central node in its functional network. This pattern suggests a top-down mechanism of action for KET. In contrast, ISO primarily stimulates subcortical regions, notably the hypothalamus, with the LC identified as the hub node within the ISO-induced functional network. This finding is consistent with a bottom-up approach (*Figure 8*; *Mashour, 2014*; *Mashour and Hudetz, 2017*). Additionally, our analysis revealed the brain nuclei that were co-activated by both KET and ISO, indicating shared neural pathways involved in general anesthesia.

Studies utilizing functional MRI, PET scans, EEG, and computational modeling have indicated that the emergence of consciousness results from active and coherent connections and communi-cation among various brain regions, enabling global information availability and integration within a neural network (*Lamme, 2006*; *Dehaene and Changeux, 2011*). Our findings revealed a significant activation throughout the cortex-dominated brain following KET administration, accompanied by a notable increase in network density (*Figure 7A*). The observed hyperactivity and heightened func-tional connectivity could potentially compromise the effective integration of information within the network, possibly leading to the loss of consciousness. This finding aligns with previous studies which indicate that KET preferentially inhibits GABAergic interneurons through NMDA receptor mediation, leading to abnormal excitatory activity in the cortex and limbic system and potentially disrupting

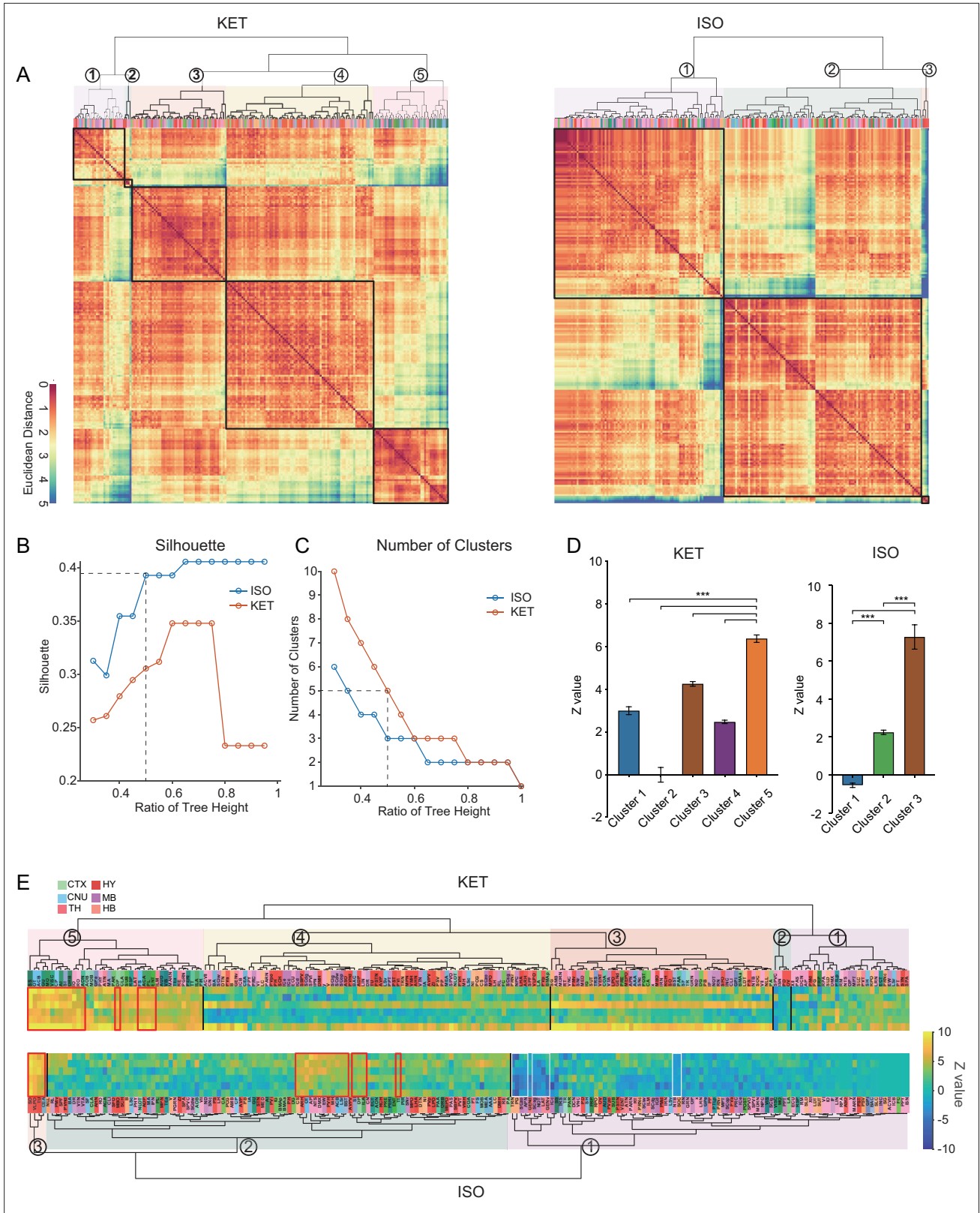

**Figure 6.** Similarities and differences in KET and ISO-activated c-Fos brain areas. (**A**) Heatmaps display hierarchical clustering of brain regions based on log-relative c-Fos density for KET and ISO groups, using complete linkage on Euclidean distances. Clusters are demarcated with a 0.5 dendrogram cut-off. (**B**) Silhouette values are plotted against the dendrogram tree height ratios in **A**. (**C**) The number of clusters identified at different dendrogram tree height ratios, with a cut-off ratio of 0.5 resulting in 5 for KET and 3 clusters for ISO (indicated by the dashed line). (**D**) Z values for identified clusters

*Figure 6 continued on next page*

*Figure 6 continued*

under KET and ISO conditions. Data are presented as mean ± SEM. One-way ANOVA with Tukey's post hoc multiple comparisons. ns: no significance; ***p<0.001. (**E**) Heatmaps illustrate Z values of log-relative c-Fos densities in brain region clusters, arranged and labeled as in **A**. Red boxes highlight regions ranking in the top 10% within their respective clusters, and white boxes indicate regions with Z values less than –2.

The online version of this article includes the following source data and figure supplement(s) for figure 6:

**Source data 1.** Hierarchical clustering of brain regions based on log-relative c-Fos density responses to KET and ISO treatments, related to *Figure 6A*.

**Source data 2.** Z values of the log relative c-Fos density in 201 brain regions in response to KET and ISO, related to *Figure 6E*.

**Figure supplement 1.** Region labels for the hierarchical clustering of the KET group in *Figure 6A*.

**Figure supplement 2.** Region labels for the hierarchical clustering of the ISO group in *Figure 6A*.

consciousness (*Seamans, 2008*; *Brown et al., 2010*). Furthermore, our analysis identified the TEa as a connector hub within this hyperactivated network (*Figure 7*). This finding, when combined with the extensive cortical activation induced by KET, support the hypothesis that consciousness alterations from KET are a result of a top-down mechanism, primarily attributed to the disruption of high-level cognitive processes regulating sensory input and consequently impairing corticothalamic circuits (*Figure 8A*; *Mashour, 2014*).

Moreover, our study employed a hierarchical clustering method to categorize brain regions by their relative activation levels, uncovering potential functional similarities and connections among these areas. Notably, during the administration of ISO, regions such as the SO, VLPO, TU, and CeA demonstrated similar and significant activation (*Figure 6E*). This aligns with previous research highlighting the involvement of SO (*Jiang-Xie et al., 2019*), VLPO (*Moore et al., 2012*), and CeA (*Hua et al., 2020*) in anesthesia mechanisms. Additionally, we noted activation in several pallidum and striatum-related brain regions under ISO (*Figure 4*), providing a new perspective for understanding the mechanism of action of ISO. Additionally, our analysis utilizing c-Fos-based functional network analysis identified the LC as a key connector hub, highlighting the importance of brainstem neurotransmitter-related nuclei during the unconscious state induced by ISO. The LC, which is the primary source of noradrenaline in the brain, plays a pivotal role in regulating vigilance, as well as respiratory and cardiovascular responses (*Magalhães et al., 2018*; *Wood and Valentino, 2017*). During ISO anesthesia, key autonomic functions are typically suppressed, including the regulation of normal body temperature, respiration, blood pressure, and heart rate (*Skovsted and Sapthavichaikul, 1977*). In this context, the activation of the LC during the administration of ISO may act as a compensatory mechanism to maintain a necessary level of autonomic response. This mechanism helps prevent the excessive suppression of crucial autonomic functions induced by ISO. While it is known that activation of the LC during ISO anesthesia can have a wake-promoting effect (*Vazey and Aston-Jones, 2014*), this phenomenon might contribute to a balance between inducing loss of consciousness and preserving essential autonomic functions during anesthesia. However, the exact role of LC activation during ISO induced anesthesia warrants further investigation. Nevertheless, this observation, combined with the significant activation of hypothalamic-related nuclei under ISO, supports the bottom-up model of unconsciousness (*Mashour and Hudetz, 2017*). This model underscores the influence of anesthetics on the hypothalamus and brainstem and focuses on how the interruption of sensory information transmission from subcortical areas to the cortex alters levels of consciousness (*Figure 8B*). Together, these findings suggest a pivotal role for the hypothalamus and brainstem regions in the loss of consciousness induced by ISO.

Identifying shared neural features between KET and ISO is essential for understanding their roles in anesthetic-induced unconsciousness, analgesia, and amnesia. The concurrent activation of the ACB and VTA alongside homeostatic control nuclei such as the LHA and MPO might suggest a shared mechanism in reward processing and homeostatic regulation (*Zhou et al., 2022*; *Al-Hasani et al., 2021*; *Mickelsen et al., 2019*; *McGinty and Szymusiak, 1990*). Additionally, the activation of neuroendocrine-related nuclei in the hypothalamus, specifically the PVH, SO, and AVPV, suggests anesthetic effects on hormonal release and homeostasis (*Gu et al., 1999*; *Adamantidis and de Lecea, 2023*). The exploration of other coactivated nuclei, including the PVT, RE, ARH, SCH, and RCH, is vital to clarify their specific anesthesia-related functions. Notably, each activated region may play multifaceted roles in anesthesia, affecting various physiological processes. The roles of these brain regions can

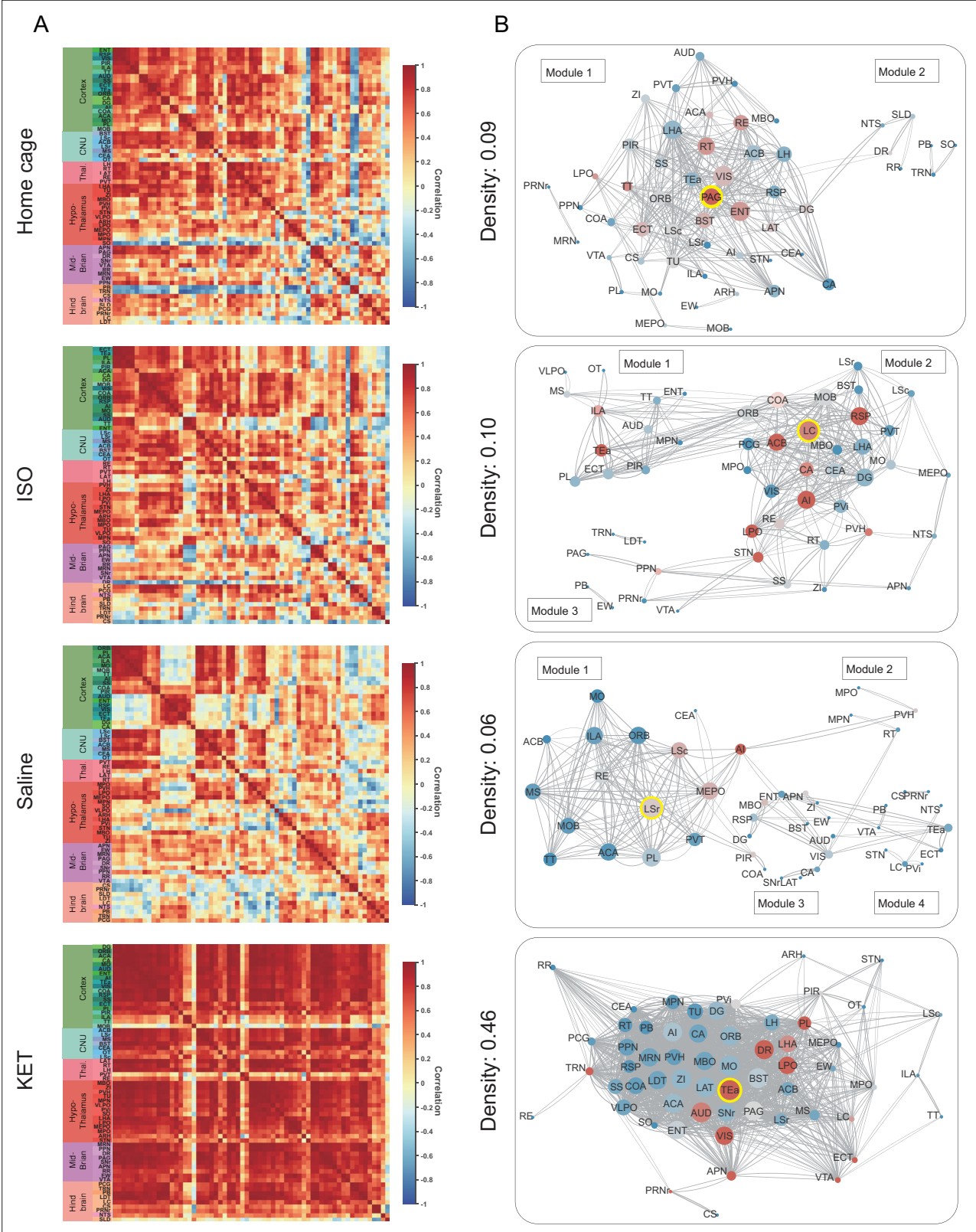

**Figure 7.** Generation of anesthetics-induced networks and identification of hub regions. (**A**) Heatmaps display the correlations of log c-Fos densities within brain regions (CTX, CNU, TH, HY, MB, and HB) for various states (home cage, ISO, saline, KET). Correlations are color-coded according to Pearson's coefficients. The brain regions within each anatomical category are organized by hierarchical clustering of their correlation coefficients. Full names and expression levels for each brain region are detailed in *Figure 3—figure supplement 1*. (**B**) Network diagrams illustrate significant positive

*Figure 7 continued on next page*

*Figure 7 continued*

correlations (p<0.05) between regions, with Pearson's r exceeding 0.82. Edge thickness indicates correlation magnitude, and node size reflects the number of connections (degree). Node color denotes betweenness centrality, with a spectrum ranging from dark blue (lowest) to dark red (highest). The networks are organized into modules consistent with the clustering depicted in *Figure 6—figure supplement 1A*.

The online version of this article includes the following source data and figure supplement(s) for figure 7:

**Source data 1.** Correlation matrix of log c-Fos density across 63 brain regions under home cage, ISO, saline, and KET conditions, related to *Figure 7A*.

**Source data 2.** Network centrality metrics for each brain region under home cage, ISO, saline, and KET conditions: degree, betweenness, and eigenvector centralities, related to *Figure 7B*.

**Figure supplement 1.** Hierarchical clustering across different conditions.

**Figure supplement 2.** Hub region characterization across different conditions: home cage (**A**), ISO (**B**), saline (**C**), and KET (**D**) treatments.

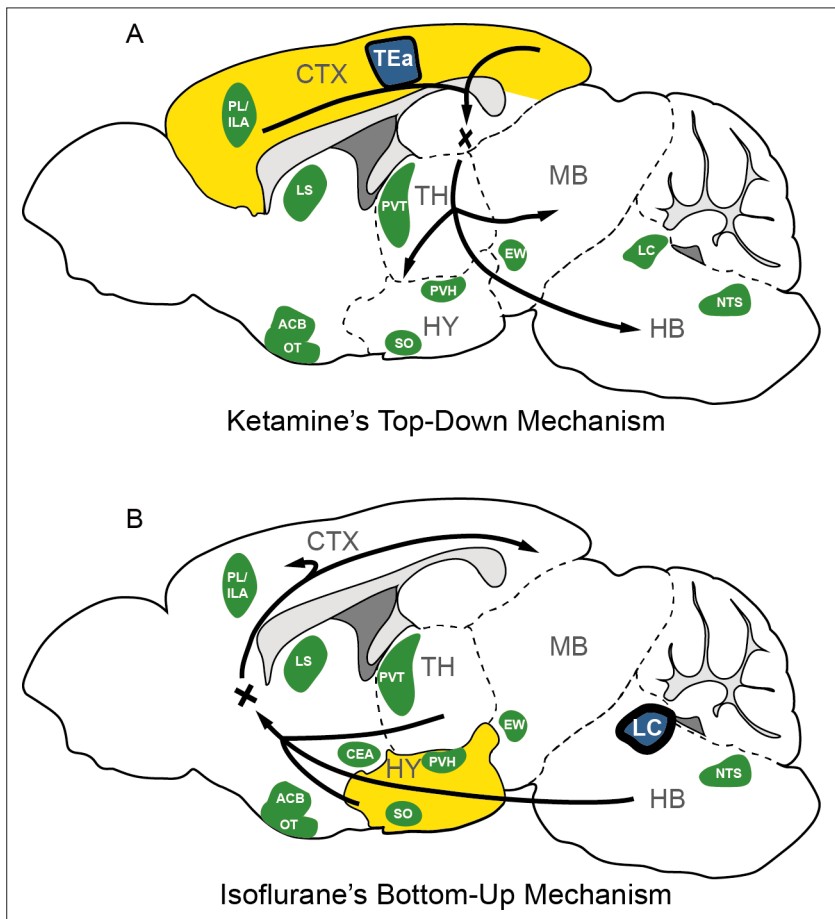

**Figure 8.** The possible mechanism for KET and ISO-induced unconsciousness. The distinct pathways of KET and ISO-induced unconsciousness can be explained by two contrasting mechanisms. (A) The 'top-down' process attributes KET's effect to widespread cortical activation (represented in yellow), with the temporal association areas (TEa) acting as the central node in the functional network (depicted in blue). (B) The 'bottom-up' approach suggests that ISO-induced unconsciousness stems from the activation of certain hypothalamic regions (highlighted in yellow), with the locus coeruleus (LC) acting as the hub node within the ISO-induced functional network. Green denotes representative regions co-activated by both ISO and KET. Adapted from Figure 1 of *Reimann and Niendorf, 2020*. PL, prelimbic area; ILA, infralimbic areas; LS, lateral septal nucleus; ACB, nucleus accumbens; OT, olfactory tubercle; PVT, paraventricular nucleus of the thalamus; EW, Edinger-Westphal nucleus; SO, supraoptic nucleus; PVH, paraventricular hypothalamic nucleus; NTS, nucleus of the solitary tract; CTX: cerebral cortex; TH: thalamus; HY, hypothalamus; MB; midbrain; HB, hindbrain.

be state-dependent and intricate, particularly in their contribution to unconsciousness, analgesia, and amnesia, underscoring the necessity for more focused research.

Our results demonstrate that c-Fos expression is significantly higher in the KET group compared to the ISO group, and in the saline group compared to the home cage group, as shown in *Figures 2 and 3*. Despite 4 days of acclimation, including handling and injections, intraperitoneal injections in the saline and KET groups might still elicit pain and stress responses in mice. This point is corroborated by the subtle yet measurable variations in brain states between the home cage and saline groups, characterized by changes in normalized EEG delta/theta power (home cage: 0.05±0.09; saline: –0.03±0.11) and EMG power (home cage: –0.37±0.34; saline: 0.04±0.13), as shown in *Figure 1—figure supplement 1*. These changes suggest a relative increase in brain activity in the saline group compared to the home cage group, potentially contributing to the higher c-Fos expression. Additionally, despite the use of consistent parameters for c-Fos labeling and imaging across all experiments, the substantial differences observed between the saline and home cage groups might be partly attributed to the fact that the operations were conducted in separate sessions. While the difference in EEG power between the ISO and home cage groups was not significant, the ISO group exhibited a similar increase in EEG power to the KET group (0.47±0.07 vs 0.59±0.10), suggesting that both agents induce loss of consciousness in mice. In terms of EMG power, ISO showed a notable decrease compared to its control group, whereas the KET group exhibited a smaller reduction (ISO: –1.815±0.10; KET: –0.96±0.21), possibly contributing to the higher c-Fos expression in the KET group. This aligns with previous research indicating that KET doses up to 150 mg/kg increase delta power and induce a wakefulness-like c-Fos expression pattern in the brain (*Lu et al., 2008*). Additionally, the differences in c-Fos expression may stem from dosages, administration routes, and distinct pharmacokinetic profiles, compounded by the absence of detailed physiological monitoring like blood pressure, heart rate, and respiration. Future research should include comprehensive physiological monitoring and controlled dosing to better understand the effects of anesthetics on brain activity.

## Materials and methods

### Animals

All animal experiments were conducted in accordance with the National Institutes of Health guidelines and were approved by the Chinese Academy of Sciences' Institute of Neuroscience (Approval No: NA-033–2022). Adult male wild-type (WT) mice (C57BL/6J; 8–10 weeks old, weight from 22 to 26 g) were purchased from institute-approved vendors (LingChang Experiment Animal Co., China). Mice were individually housed and maintained on a 12 hr:12 hr light/dark cycle (lights on at 07:00 a.m. and off at 07:00 p.m.) with food and water available ad libitum.

### Drug administration

All experiments were conducted between 13:00 and 14:30 (ZT6–ZT7.5). We adapted mice to handling and the anesthesia chamber (10×15 × 15 cm) for 4 days to minimize experimental confound-induced c-Fos induction. For KET administration, adult male mice were handled for 10 min/day, and normal saline (NS) was injected intraperitoneally (i.p.) for 4 consecutive days at 13:00. On day 5, a randomly chosen mouse received an injection of ketamine (Gutian Medicine, H35020148) (n=6), and the control groups (n=8) received the same volume of saline. ISO group (n=6) mice were handled and inhaled 1.5% isoflurane (RWD Life Science, 1903715) at 13:00 on day 4 in the chamber. Meanwhile, the control groups (n=6) were left undisturbed in their home cages before sampling. We confirmed the loss of righting reflex at 5 min after anesthetic exposure. For 90 min after KET injection or ISO inhalation, mice were deeply anesthetized with 5% ISO and transcardially perfused with 35 ml 0.1 M phosphate-buffered saline (PBS) followed by 35 ml 4% paraformaldehyde (PFA). The brains were then removed and postfixed overnight with 4% PFA. Following fixation, the brains were dehydrated for 48 hr with 30% sucrose (wt/vol) in PBS. Coronal sections (50 µm) of the whole brain were cut using a cryostat (HM525 NX, Thermo Fisher Scientific) after being embedded with OCT compound (NEG-50, Thermo Fisher Scientific) and freezing.

## Anesthesia depth measurement

EEG and EMG recordings were conducted as previously described (*Luo et al., 2023*). Specifically, a stainless-steel electrode was positioned over the frontal cortex for EEG measurements, while an insulated EMG electrode was inserted into the neck muscles. Additionally, a reference electrode was placed above the cerebellum. The EEG/EMG data were captured using TDT system-3 amplifiers, specifically RZ5 +RA16 PA and RZ2 +PZ5 configurations. To measure the depth of anesthesia using EEG/EMG signals, a fast Fourier transform spectral analysis was employed, featuring a frequency resolution of 0.18 Hz. Data was noted every 5 s through a MATLAB tool and subsequently verified manually by experts. The assessment was based on EMG power and the ratio of EEG's delta power (0.5–4 Hz) to theta power (6–10 Hz). EEG and EMG power values within 30 min post-administration were normalized to a 5-min pre-administration baseline.

## Immunohistochemistry

One out of every three brain slices (100 μm intervals) of each whole brain was washed three times with 0.1 M phosphate-buffered saline (PBS) for 10 min and permeabilized for 30 min at room temperature with 0.3% Triton X-100 in PBS (PBST). Slices were incubated for 2 hr at room temperature with 2% normal donkey serum (Sigma, G6767) in PBS overnight at 4 °C with c-Fos primary antibodies (Synaptic Systems, 226003, RRID:AB-2231974; 1:500) diluted in PBS with 1% donkey serum. After three washes with PBST, slices were incubated with the Cy3 donkey anti-rabbit (Jackson Immuno Research, 711-165-152; RRID: AB-2307443; 1:200) secondary antibody for 2 hr at room temperature. Immunostained brain slices were first mounted using VECTASHIELD mounting medium containing DAPI. Comprehensive imaging of these slices was initially performed with a 10×objective on a fluorescent microscope (VS120, Olympus). Regions showing statistically significant differences were then imaged in greater detail using a 20×objective on a confocal microscope (FV300, Olympus). Due to the limited scope of a single field of view, either 2 or 4 adjacent fields of view were stitched together to offer a comprehensive representation of specific brain regions, such as the PL, ILA, LSc, LSr, SS, and VISC.

## Quantification of c-Fos-positive cells

The procedures used for c-Fos analysis were based on previous research (*Ma et al., 2021*). A custom-written software package was employed for cellular quantification within brain images. The software consists of three components: atlas rotation, image registration, and signal detection.

### Atlas rotation module

Utilizing the Allen Mouse Brain Atlas, this module allows rotations in both horizontal and vertical dimensions to align with mouse brain samples. To determine the appropriate rotation angles, we manually pinpointed anatomical landmarks corresponding to the samples. For the left-right axis rotation, we chose points from the CA3 pyramidal layer in both hemispheres and the posterior slice of the dentate gyrus, specifically the granule cell layer. For the dorsal-ventral axis rotation, we identified key anatomical landmarks. These include the initial connections of the anterior commissure and corpus callosum between the hemispheres, as well as ventral regions like the interpeduncular nucleus and the suprachiasmatic nucleus. After determining these rotation angles, we adjusted the reference atlas to match our samples.

### Image registration module

This module uses a tool that aligns brain slice images with a reference atlas, facilitating the alignment of overall brain region distribution. Registration starts by matching the coronal plane of the sample sections to the atlas. After defining the external boundaries of the brain section, the system performs geometric transformations on the section to optimize its fit with the atlas.

### Signal detection module

The detection module is specifically designed to automatically label c-Fos-positive cells. Following detection, each digitized brain section image underwent manual verification to ensure the accuracy and precision of the c-Fos-positive cell labeling.

## Hierarchical clustering

Prior to hierarchical clustering in *Figures 5 and 6*, we calculated the relative c-Fos densities by dividing the c-Fos densities of each brain region in the experimental groups by their respective controls and then performed a log transformation on these values to obtain the log relative c-Fos densities. These log ratios, which normalize the data and reduce variance, indicate differential expression with a value of zero denoting no change compared to control. After normalizing the data, we performed hierarchical clustering by first computing the pairwise Euclidean distances among brain regions. Regions with the smallest distances, indicating high similarity, were grouped iteratively. Cluster boundaries were defined using complete linkage, ensuring homogeneity within clusters by considering the largest distance between members. In *Figure 7A*, hierarchical clustering was performed within each of the cerebral cortex (CTX), thalamus (TH), cerebral nuclei (CNU), hypothalamus (HY), midbrain (MB), and hindbrain (HB) regions based on the log-transformed c-Fos density correlations. In *Figure 7—figure supplement 1*, hierarchical clustering was performed based on the interregional log c-Fos density correlations to identify modules of coactivation within each treatment group, revealing underlying functional connectivity networks (*Kimbrough et al., 2020*). To determine the statistical significance of c-Fos expression differences, we computed the Z value for each treatment condition—KET and ISO—by dividing the log relative c-Fos density by the standard error. Positive Z values indicate higher values than the control, and negative Z values indicate lower values than the control.

## Network generation

To evaluate how functional connectivity changed under general anesthesia in WT mice, we extracted 63 brain regions from major brain subdivisions (CTX, CNU, TH, HY, MB, and HB) listed in *Figure 7A*. Correlation matrices were generated by computing Pearson correlation coefficients from interregional c-Fos expression in the 63 regions. Mean correlations were calculated to assess changes in functional connectivity between these major subdivisions of the brain. Weighted undirected networks were constructed by considering correlations with Pearson's $r \geq 0.82$, corresponding to a one-tailed significance level of $p < 0.05$ (uncorrected). The nodes in the networks represent brain regions, and the correlations that survived thresholding were considered connections. Theoretical graph analysis was performed using Brain Connectivity Toolbox (https://sites.google.com/site/bctnet/, version 2019-03-03) in MATLAB R2021 (The MathWorks Inc) (*Rubinov and Sporns, 2010*). Network visualization was performed using Cytoscape (version 3.2.1) (*Shannon et al., 2003*).

## Hub identification

Network centrality was evaluated using degree, betweenness, and eigenvector centrality measures to identify potential hub regions (*Sciarra et al., 2018*). Degree counts the number of edges connected to a node, signifying its immediate influence. Betweenness centrality is gauged by the number of shortest paths passing through a node, indicating its role as a connector within the network. Eigenvector centrality measures a node's influence by the centrality of its connections, valuing nodes linked to well-connected neighbors. High eigenvector centrality indicates significant influence through these high-quality connections within the network.

## Statistical analysis

Sample size was determined based on prior studies (*Lu et al., 2008*; *Yatziv et al., 2020*). For normally distributed datasets, one-sample t-tests and one-way ANOVA with Tukey's post hoc analysis were used for single-sample and multi-group comparisons, respectively. For non-normal data, the Wilcoxon signed-rank and Kruskal-Wallis tests, followed by Dunn's test, identified disparities in single-sample and multi-group contexts. In *Figure 2* and *Figure 2—figure supplement 1*, c-Fos densities in both experimental and control groups were log-transformed. Z-scores were calculated for each brain region by normalizing these log-transformed values against the mean and standard deviation of its respective control group. This involved subtracting the control mean from the experimental value and dividing the result by the control standard deviation. For statistical analysis, Z-scores were compared to a null distribution with a zero mean, and adjustments were made for multiple comparisons using the Benjamini–Hochberg method with a 5% false discovery rate (Q) (*Benjamini and Hochberg, 1995*). Pearson correlation coefficients (R) were transformed into Z-scores using Fisher's Z transformation before computing group means and making statistical comparisons in *Figure 7—figure supplement*

*1D*. All statistical analyses were conducted using GraphPad Prism 9.0 (GraphPad Software, USA) and MATLAB R2021 (Mathworks Inc).

## Acknowledgements

We express our gratitude to our interns, Chuhang Wong from Imperial College London and Jiale Huang from ShanghaiTech University, for their assistance in cell counting.

## Additional information

### Funding

| Funder | Grant reference number | Author |
|---|---|---|
| National Natural Science Foundation of China | 82271292 | Yingwei Wang |
| National Natural Science Foundation of China | 81730031 | Yingwei Wang |
| National Natural Science Foundation of China | 82371286 | Mengqiang Luo |
| National Natural Science Foundation of China | 82101350 | Mengqiang Luo |
| Shanghai Municipal Key Clinical Specialty | shslczdzk06901 | Yingwei Wang |
| Foundation of Shanghai Municipal Science and Technology Medical Innovation Research Project | 23Y21900600 | Yingwei Wang |

The funders had no role in study design, data collection and interpretation, or the decision to submit the work for publication.

### Author contributions

Yue Hu, Conceptualization, Formal analysis, Investigation, Visualization, Methodology, Writing - original draft, Writing - review and editing; Wenjie Du, Conceptualization, Formal analysis, Visualization, Writing - original draft, Writing - review and editing; Jiangtao Qi, Conceptualization, Software, Formal analysis; Huoqing Luo, Formal analysis, Methodology, Writing - review and editing; Zhao Zhang, Resources, Supervision, Investigation, Writing - review and editing; Mengqiang Luo, Resources, Supervision, Funding acquisition, Project administration, Writing - review and editing; Yingwei Wang, Conceptualization, Resources, Supervision, Funding acquisition, Project administration, Writing - review and editing

### Author ORCIDs

Yue Hu http://orcid.org/0009-0009-2806-2557
Yingwei Wang http://orcid.org/0000-0003-1633-8834

### Ethics

All animal experiments were conducted in accordance with the National Institutes of Health guidelines and were approved by the Chinese Academy of Sciences' Institute of Neuroscience (Approval No: NA-033-2022). This study strictly adhered to the principles of care and use of laboratory animals. Appropriate anesthetic and analgesic measures were employed prior to any procedures to minimize discomfort and pain to the animals to the greatest extent possible.

Reviewer #2 (Public Review): https://doi.org/10.7554/eLife.88420.5.sa1
Reviewer #3 (Public Review): https://doi.org/10.7554/eLife.88420.5.sa2
Author Response https://doi.org/10.7554/eLife.88420.5.sa3

## Additional files

### Supplementary files
• MDAR checklist

### Data availability
The dataset supporting the findings of this study is available in the Harvard Dataverse repository. Custom scripts for c-Fos expression analysis are available on GitHub (copy archived at *YWWLab, 2024*). Source data files for Figures 2, 3, 4, 5, 6, and 7 have been provided.

The following dataset was generated:

| Author(s) | Year | Dataset title | Dataset URL | Database and Identifier |
|---|---|---|---|---|
| Wang Y | 2024 | Replication Data for: Comparative brain-wide mapping of ketamine- and isoflurane-activated nuclei and functional networks in the mouse brain | https://doi.org/10.7910/DVN/ZZONDL | Harvard Dataverse, 10.7910/DVN/ZZONDL |

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
