## [Editor Report · eLife assessment]

This **important** study used single-cell whole-brain imaging of the immediate early gene Fos to identify the brain areas recruited by two anesthetics, ketamine and isoflurane. The utilization of a custom software package to align and analyze brain images for c-Fos positive cells stands out as an impressive component of the approach. The results provide **solid** evidence that these anesthetics might induce anesthesia via different brain regions and pathways, and raw fos showed shared and distinct activation patterns after ketamine- v. isoflurane-based anesthesia. Though differences could also be due, as the authors note, to differences in dose and route of administration. This paper may be of interest to preclinical and clinical scientists working with anesthetic and dissociative drugs.

---

## [Referee Report · Reviewer #2 (Public Review)]

Summary: In the revised manuscript, the authors aim to investigate brain-wide activation patterns following administration of the anesthetics ketamine and isoflurane, and conduct comparative analysis of these patterns to understand shared and distinct mechanisms of these two anesthetics. To this end, they perform Fos immunohistochemistry in perfused brain sections to label active nuclei, use a custom pipeline to register images to the ABA framework and quantify Fos+ nuclei, and perform multiple complementary analyses to compare activation patterns across groups.

In the latest revision, I am happy to say that the authors have greatly improved their manuscript. The data are now well analyzed and the experiments fully described. They addressed all of my concerns. It is an interesting study.

---

## [Referee Report · Reviewer #3 (Public Review)]

The present study presents a comprehensive exploration of the distinct impacts of Isoflurane and Ketamine on c-Fos expression throughout the brain. To understand the varying responses across individual brain regions to each anesthetic, the researchers employ hierarchical clustering and c-Fos-based functional network analysis. The methodology employed in this research is both methodical and expansive. Notably, the utilization of a custom software package to align and analyze brain images for c-Fos positive cells stands out as an impressive addition to their approach. This innovative technique enables effective quantification of neural activity and enhances our understanding of how anesthetic drugs influence brain networks as a whole.

The primary novelty of this paper lies in the comparative analysis of two anesthetics, Ketamine and Isoflurane, and their respective impacts on brain-wide c-Fos expression. The study reveals the distinct pathways through which these anesthetics induce loss of consciousness. Ketamine primarily influences the cerebral cortex, while Isoflurane targets subcortical brain regions. This finding highlights the differing mechanisms of action employed by these two anesthetics-a top-down approach for Ketamine and a bottom-up mechanism for Isoflurane. Furthermore, this study uncovers commonly activated brain regions under both anesthetics, advancing our knowledge about the mechanisms underlying general anesthesia.

---

## [Author Response]

The following is the authors’ response to the previous reviews.

**Reviewer #2 (Public Review):**
Summary:In the revised manuscript, the authors aim to investigate brain-wide activation patterns following administration of the anesthetics ketamine and isoflurane, and conduct comparative analysis of these patterns to understand shared and distinct mechanisms of these two anesthetics. To this end, they perform Fos immunohistochemistry in perfused brain sections to label active nuclei, use a custom pipeline to register images to the ABA framework and quantify Fos+ nuclei, and perform multiple complementary analyses to compare activation patterns across groups.In the latest revision, the authors have made some changes in response to our previous comments on how to fix the analyses. However, the revised analyses were not changed correctly and remain flawed in several fundamental ways.Critical problems:(1) Before one can perform higher level analyses such as hiearchal cluster or network hub (or PC) analysis, it is fundamental to validate that you have significant differences of the raw Fos expression values in the first place. First of all, this means showing figures with the raw data (Fos expression levels) in some form in Figures 2 and 3 before showing the higher level analyses in Figures 4 and 5; this is currently switched around. Second and most importantly, when you have a large number of brain areas with large differences in mean values and variance, you need to account for this in a meaningful way. Changing to log values is a step in the right direction for mean values but does not account well for differences in variance. Indeed, considering the large variances in brain areas with high mean values and variance, it is a little difficult to believe that all brain regions, especially brain areas with low mean values, passed corrections for multiple comparisons test. We suggested Z-scores relative to control values for each brain region; this would have accounted for wide differences in mean values and variance, but this was not done. Overall, validation of anesthesia-induced differences in Fos expression levels is not yet shown.

(a) Reordering the figures.

Thank you for your suggestion. We have added Figure 2 (for 201 brain regions) and Figure 2—figure supplement 1 (for 53 brain regions) to demonstrate the statistical differences in raw Fos expression between KET and ISO compared to their respective control groups. These figures specifically present the raw c-Fos expression levels for both KET and ISO in the same brain areas, providing a fundamental basis for the subsequent analyses. Additionally, we have moved the original Figures 4 and 5 to Figures 3 and 4.

(b) Z-score transformation and validation of anesthesia-induced differences in Fos expression.

Thank you for your suggestion. Before multiple comparisons, we transformed the data into log c-Fos density and then performed Z-scores relative to control values for each brain region. Indeed, through Z-score transformation, we have identified a larger number of significantly activated brain regions in Figure 2. The number of brain regions showing significant activation increased by 100 for KET and by 39 for ISO. We have accordingly updated the results section to include these findings in Line 80-181. Besides, we have added the following content in the Statistical Analysis section in Line 489: "…In Figure 2 and Figure 2–figure supplement 1, c-Fos densities in both experimental and control groups were log-transformed. Z-scores were calculated for each brain region by normalizing these log-transformed values against the mean and standard deviation of its respective control group. This involved subtracting the control mean from the experimental value and dividing the result by the control standard deviation. For statistical analysis, Z-scores were compared to a null distribution with a zero mean, and adjustments were made for multiple comparisons using the Benjamini–Hochberg method with a 5% false discovery rate (Q)..…".

**Author response image 1. sa3fig1:** KET and ISO induced c-Fos expression relative to their respective control group across 201 distinct brain regions. Z-scores represent the normalized c-Fos expression in the KET and ISO groups, calculated against the mean and standard deviation from their respective control groups. Statistical analysis involved the comparison of Z-scores to a null distribution with a zero mean and adjustment for multiple comparisons using the Benjamini–Hochberg method at a 5% false discovery rate (*p < 0.05, **p < 0.01, ***p < 0.001). n = 6, 6, 8, 6 for the home cage, ISO, saline, and KET, respectively. Missing values resulted from zero standard deviations in control groups. Brain regions are categorized into major anatomical subdivisions, as shown on the left side of the graph.

**Author response image 2. sa3fig2:** KET and ISO induced c-Fos expression relative to their respective control group across 53 distinct brain regions. Z-scores for c-Fos expression in the KET and ISO groups were normalized to the mean and standard deviation of their respective control groups. Statistical analysis involved the comparison of Z-scores to a null distribution with a zero mean and adjustment for multiple comparisons using the Benjamini–Hochberg method at a 5% false discovery rate (*p < 0.05, **p < 0.01, ***p < 0.001). Brain regions are organized into major anatomical subdivisions, as indicated on the left side of the graph.

(2) Let's assume for a moment that the raw Fos expression analyses indicate significant differences. They used hierarchal cluster analyses as a rationale for examining 53 brain areas in all subsequent analyses of Fos expression following isoflurane versus home cage or ketamine versus saline. Instead, the authors changed to 201 brain areas with no validated rationale other than effectively saying 'we wanted to look at more brain areas'. And then later, when they examined raw Fos expression values in Figures 4 and 5, they assess 43 brain areas for ketamine and 20 brain areas for isoflurane, without any rationale for why choosing these numbers of brain areas. This is a particularly big problem when they are trying to compare effects of isoflurane versus ketamine on Fos expression in these brain areas - they did not compare the same brain areas.

(a) Changing to 201 brain areas with validated rationale.

Thank you for your question. We have revised the original text from “To enhance our analysis of c-Fos expression patterns induced by KET and ISO, we expanded our study to 201 subregions.” to Line 100: "…To enable a more detailed examination and facilitate clearer differentiation and comparison of the effects caused by KET and ISO, we subdivided the 53 brain regions into 201 distinct areas. This approach, guided by the standard mouse atlas available at http://atlas.brain-map.org/atlas, allowed for an in-depth analysis of the responses in various brain regions…". For hierarchal cluster analyses from 53 to 201 brain regions, Line 215: "…To achieve a more granular analysis and better discern the responses between KET and ISO, we expanded our study from the initial 53 brain regions to 201 distinct subregions…"

(b) Compare the same brain areas for KET and ISO and the rationale for why choosing these numbers of brain areas in Figures 3 and 4.

We apologize for the confusion and lack of clarity regarding the selection of brain regions for analysis. In Figure 2 and Figure 2—figure supplement 1, we display the c-Fos expression in the same brain regions affected by KET and ISO. In Figures 3 and 4, we applied a uniform standard to specifically report the brain areas most prominently activated by KET and ISO, respectively. As specified in Line 104: "…Compared to the saline group, KET activated 141 out of a total of 201 brain regions (Figure 2). To further identify the brain regions that are most significantly affected by KET, we calculated Cohen's d for each region to quantify the magnitude of activation and subsequently focused on those regions that had a corrected p-value below 0.05 and effect size in the top 40% (Figure 3, Figure 3—figure supplement 1)…" and Line 142: "…Using the same criteria applied to KET, which involved selecting regions with Cohen's d values in the top 40% of significantly activated areas from Figure 2, we identified 32 key brain regions impacted by ISO (Figure 4, Figure 4—figure supplement 1).…".

Moreover, we illustrate the co-activated brain regions by KET and ISO in Figure 4C. As detailed in Lines 167-180:"…The co-activation of multiple brain regions by KET and ISO indicates that they have overlapping effects on brain functions. Examples of these effects include impacts on sensory processing, as evidenced by the activation of the PIR, ENT 1, and OT2, pointing to changes in sensory perception typical of anesthetics. Memory and cognitive functions are influenced, as indicated by the activation of the subiculum (SUB) 3, dentate gyrus (DG) 4, and RE 5. The reward and motivational systems are engaged, involving the ACB and ventral tegmental area (VTA), signaling the modulation of reward pathways 6. Autonomic and homeostatic control are also affected, as shown by areas like the lateral hypothalamic area (LHA) 7 and medial preoptic area (MPO) 8, emphasizing effects on functions such as feeding and thermoregulation. Stress and arousal responses are impacted through the activation of the paraventricular hypothalamic nucleus (PVH) 10,11 and LC 12. This broad activation pattern highlights the overlap in drug effects and the complexity of brain networks in anesthesia…". Below are the revised Figures 3 and 4.

(1) Chapuis, J. et al. Lateral entorhinal modulation of piriform cortical activity and fine odor discrimination. J. Neurosci. 33, 13449-13459 (2013). https://doi.org:10.1523/jneurosci.1387-13.2013

(2) Giessel, A. J. & Datta, S. R. Olfactory maps, circuits and computations. Curr. Opin. Neurobiol. 24, 120-132 (2014). https://doi.org:10.1016/j.conb.2013.09.010

(3) Roy, D. S. et al. Distinct Neural Circuits for the Formation and Retrieval of Episodic Memories. Cell 170, 1000-1012.e1019 (2017). https://doi.org:10.1016/j.cell.2017.07.013

(4) Sun, X. et al. Functionally Distinct Neuronal Ensembles within the Memory Engram. Cell 181, 410-423.e417 (2020). https://doi.org:10.1016/j.cell.2020.02.055

(5) Huang, X. et al. A Visual Circuit Related to the Nucleus Reuniens for the Spatial-Memory-Promoting Effects of Light Treatment. Neuron (2021).

(6) Al-Hasani, R. et al. Ventral tegmental area GABAergic inhibition of cholinergic interneurons in the ventral nucleus accumbens shell promotes reward reinforcement. Nat. Neurosci. 24, 1414-1428 (2021). https://doi.org:10.1038/s41593-021-00898-2

(7) Mickelsen, L. E. et al. Single-cell transcriptomic analysis of the lateral hypothalamic area reveals molecularly distinct populations of inhibitory and excitatory neurons. Nat. Neurosci. 22, 642-656 (2019). https://doi.org:10.1038/s41593-019-0349-8

(8) McGinty, D. & Szymusiak, R. Keeping cool: a hypothesis about the mechanisms and functions of slow-wave sleep. Trends Neurosci. 13, 480-487 (1990). https://doi.org:10.1016/0166-2236(90)90081-k

(9) Mullican, S. E. et al. GFRAL is the receptor for GDF15 and the ligand promotes weight loss in mice and nonhuman primates. Nat. Med. 23, 1150-1157 (2017). https://doi.org:10.1038/nm.4392

(10) Rasiah, N. P., Loewen, S. P. & Bains, J. S. Windows into stress: a glimpse at emerging roles for CRH(PVN) neurons. Physiol. Rev. 103, 1667-1691 (2023). https://doi.org:10.1152/physrev.00056.2021

(11) Islam, M. T. et al. Vasopressin neurons in the paraventricular hypothalamus promote wakefulness via lateral hypothalamic orexin neurons. Curr. Biol. 32, 3871-3885.e3874 (2022). https://doi.org:10.1016/j.cub.2022.07.020

(12) Ross, J. A. & Van Bockstaele, E. J. The Locus Coeruleus- Norepinephrine System in Stress and Arousal: Unraveling Historical, Current, and Future Perspectives. Front Psychiatry 11, 601519 (2020). https://doi.org:10.3389/fpsyt.2020.601519

**Author response image 3. sa3fig3:** Brain regions exhibiting significant activation by KET. (**A**) Fifty-five brain regions exhibited significant KET activation. These were chosen from the 201 regions analyzed in Figure 2, focusing on the top 40% ranked by effect size among those with corrected p values less than 0.05. Data are presented as mean ± SEM, with p-values adjusted for multiple comparisons (*p < 0.05, **p < 0.01, ***p < 0.001). (**B**) Representative immunohistochemical staining of brain regions identified in Figure 3A, with control group staining available in Figure 3—figure supplement 1. Scale bar: 200 µm.

**Author response image 4. sa3fig4:** Brain regions exhibiting significant activation by ISO. (**A**) Brain regions significantly activated by ISO were initially identified using a corrected p-value below 0.05. From these, the top 40% in effect size (Cohen’s d) were further selected, resulting in 32 key areas. p-values are adjusted for multiple comparisons (**p < 0.01, ***p < 0.001). (**B**) Representative immunohistochemical staining of brain regions identified in Figure 4A. Control group staining is available in Figure 4—figure supplement 1. Scale bar: 200 µm. Scale bar: 200 µm. (**C**) A Venn diagram displays 43 brain regions co-activated by KET and ISO, identified by the adjusted p-values (p < 0.05) for both KET and ISO. CTX: cerebral cortex; CNU: cerebral nuclei; TH: thalamus; HY: hypothalamus; MB: midbrain; HB: hindbrain.

Brain regions exhibiting significant activation by ISO. (A) Brain regions significantly activated by ISO were initially identified using a corrected p-value below 0.05. From these, the top 40% in effect size (Cohen’s d) were further selected, resulting in 32 key areas. p-values are adjusted for multiple comparisons (**p < 0.01, ***p < 0.001). (B) Representative immunohistochemical staining of brain regions identified in Figure 4A. Control group staining is available in Figure 4—figure supplement 1. Scale bar: 200 µm. Scale bar: 200 µm. (C) A Venn diagram displays 43 brain regions co-activated by KET and ISO, identified by the adjusted p-values (p < 0.05) for both KET and ISO. CTX: cerebral cortex; CNU: cerebral nuclei; TH: thalamus; HY: hypothalamus; MB: midbrain; HB: hindbrain.

Less critical comments:(3) The explanation of hierarchical level's in lines 90-95 did not make sense.

We have revised the section that initially stated in lines 90-95, "…Based on the standard mouse atlas available at http://atlas.brain-map.org/, the mouse brain was segmented into nine hierarchical levels, totaling 984 regions. The primary level consists of grey matter, the secondary of the cerebrum, brainstem, and cerebellum, and the tertiary includes regions like the cerebral cortex and cerebellar nuclei, among others, with some regions extending to the 8th and 9th levels. The fifth level comprises 53 subregions, with detailed expression levels and their respective abbreviations presented in Supplementary Figure 2…". Our revised description, now in line 91: "…Building upon the framework established in previous literature, our study categorizes the mouse brain into 53 distinct subregions1…"

(1) Do JP, Xu M, Lee SH, Chang WC, Zhang S, Chung S, Yung TJ, Fan JL, Miyamichi K, Luo L et al: Cell type-specific long-range connections of basal forebrain circuit. Elife 2016, 5.

(4) I am still perplexed by why the authors consider the prelimbic and infralimbic cortex 'neuroendocrine' brain areas in the abstract. In contrast, the prelimbic and infralimbic were described better in the introduction as "associated information processing" areas.

Thank you for bringing this to our attention. We agree that classifying the prelimbic and infralimbic cortex as 'neuroendocrine' in the abstract was incorrect, which was an oversight on our part. In the revised version, as detailed in line 167, we observed an increased number of brain regions showing overlapping activation by both KET and ISO, which is depicted in Figure 4C. This extensive co-activation across various regions makes it challenging to narrowly define the functional classification of each area. Consequently, we have revised the abstract, updating this in line 21: "…KET and ISO both activate brain areas involved in sensory processing, memory and cognition, reward and motivation, as well as autonomic and homeostatic control, highlighting their shared effects on various neural pathways.…".

(5) It looks like overall Fos levels in the control group Home (ISO) are a magnitude (~10-fold) lower than those in the control group Saline (KET) across all regions shown. This large difference seems unlikely to be due to a biologically driven effect and seems more likely to be due to a technical issue, such as differences in staining or imaging between experiments. The authors discuss this issue but did not answer whether the Homecage-ISO experiment or at least the Fos labeling and imaging performed at the same time as for the Saline-Ketamine experiment?

Thank you for highlighting this important point. The c-Fos labeling and imaging for the Home (ISO) and Saline (KET) groups were carried out in separate sessions due to the extensive workload involved in these processes. This study processed a total of 26 brain samples. Sectioning the entire brain of each mouse required approximately 3 hours, yielding 5 slides, with each slide containing 12 to 16 brain sections. We were able to stain and image up to 20 slides simultaneously, typically comprising 2 experimental groups and 2 corresponding control groups. Imaging these 20 slides at 10x magnification took roughly 7 hours, while additional time was required for confocal imaging of specific areas of interest at 20x magnification. Given the complexity of these procedures, to ensure consistency across all experiments, they were conducted under uniform conditions. This included the use of consistent primary and secondary antibody concentrations, incubation times, and imaging parameters such as fixed light intensity and exposure time. Furthermore, in the saline and KET groups, intraperitoneal injections might have evoked pain and stress responses in mice despite four days of pre-experiment acclimation, which could have contributed to the increased c-Fos expression observed. This aspect, along with the fact that procedures were conducted in separate sessions, might have introduced some variations. Thus, we have included a note in our discussion section in Line 353: "…Despite four days of acclimation, including handling and injections, intraperitoneal injections in the saline and KET groups might still elicit pain and stress responses in mice. This point is corroborated by the subtle yet measurable variations in brain states between the home cage and saline groups, characterized by changes in normalized EEG delta/theta power (home cage: 0.05±0.09; saline: -0.03±0.11) and EMG power (home cage: -0.37±0.34; saline: 0.04±0.13), as shown in Figure 1–figure supplement 1. These changes suggest a relative increase in brain activity in the saline group compared to the home cage group, potentially contributing to the higher c-Fos expression. Additionally, despite the use of consistent parameters for c-Fos labeling and imaging across all experiments, the substantial differences observed between the saline and home cage groups might be partly attributed to the fact that the operations were conducted in separate sessions.…"

**Reviewer #3 (Public Review):**
The present study presents a comprehensive exploration of the distinct impacts of Isoflurane and Ketamine on c-Fos expression throughout the brain. To understand the varying responses across individual brain regions to each anesthetic, the researchers employ principal component analysis (PCA) and c-Fos-based functional network analysis. The methodology employed in this research is both methodical and expansive. Notably, the utilization of a custom software package to align and analyze brain images for c-Fos positive cells stands out as an impressive addition to their approach. This innovative technique enables effective quantification of neural activity and enhances our understanding of how anesthetic drugs influence brain networks as a whole.The primary novelty of this paper lies in the comparative analysis of two anesthetics, Ketamine and Isoflurane, and their respective impacts on brain-wide c-Fos expression. The study reveals the distinct pathways through which these anesthetics induce loss of consciousness. Ketamine primarily influences the cerebral cortex, while Isoflurane targets subcortical brain regions. This finding highlights the differing mechanisms of action employed by these two anesthetics-a top-down approach for Ketamine and a bottom-up mechanism for Isoflurane. Furthermore, this study uncovers commonly activated brain regions under both anesthetics, advancing our knowledge about the mechanisms underlying general anesthesia.

We are thankful for your positive and insightful comments on our study. Your recognition of the study's methodology and its significance in advancing our understanding of anesthetic mechanisms is greatly valued. By comprehensively mapping c-Fos expression across a wide range of brain regions, our study reveals the distinct and overlapping impacts of these anesthetics on various brain functions, providing a valuable foundation for future research into the mechanisms of general anesthesia, potentially guiding the development of more targeted anesthetic agents and therapeutic strategies. Thus, we are confident that our work will captivate the interest of our readers.